# Measuring and explaining inequality of continuous care for people living with HIV receiving antiretroviral therapy in Kunming, China

Yongmei Jin[1,2], Sawitri Assanangkornchai[2]*, Yingrong Du[1], Jun Liu[1], Jingsong Bai[1], Yongrui Yang[1]

**1** Department of Infectious Diseases, The Third People's Hospital of Kunming City, Kunming, Yunnan, People's Republic of China, **2** Epidemiology Unit, Faculty of Medicine, Prince of Songkla University, Hat Yai, Songkhla, Thailand

* savitree.a@psu.ac.th

## Abstract

### Background

In the context of scaling up free antiretroviral therapy (ART), healthcare equality is essential for people living with HIV. We aimed to assess socioeconomic-related inequalities in uptake of continuous care for people living with HIV receiving ART, including retention in care in the last six months, routine toxicity monitoring, adequate immunological and virological monitoring, and uptake of mental health assessment in the last 12 months. We also determined the contributions of socioeconomic factors to the degree of inequalities.

### Methods

A hospital-based cross-sectional survey was conducted among consecutive clients visiting an HIV treatment center in Kunming, China in 2019. Participants were 702 people living with HIV aged ≥18 years (median age: 41.0 years, 69.4% male) who had been on ART for 1–5 years. Socioeconomic-related inequality and its contributing factors were assessed by a normalized concentration index ($CI_n$) with a decomposition approach.

### Results

The uptake of mental health assessment was low (15%) but significantly higher among the rich ($CI_n$ 0.1337, 95% CI: 0.0140, 0.2534). Retention in care, toxicity, and immunological monitoring were over 80% but non-significant in favor of the rich ($CI_n$: 0.0117, 0.0315, 0.0736, respectively). The uptake of adequate virological monitoring was 15% and higher among the poor ($CI_n$ = -0.0308). Socioeconomic status positively contributed to inequalities of all care indicators, with the highest contribution for mental health assessment (124.9%) and lowest for virological monitoring (2.7%).

**Data Availability Statement:** All relevant data are within the manuscript and its Supporting information files.

**Funding:** This work was supported by the Higher Education Research Promotion and Thailand's Education Hub for Southern Region of ASEAN Countries Project Office of the Higher Education Commission and the Scientific Research Fund of Yunnan Provincial Department of Education, Yunnan province, China (Grant No. 2020J0610).

**Competing interests:** The authors have declared that no competing interests exist.

## Conclusions

These findings suggest virological monitoring and mental health assessment be given more attention in long-term HIV care. Policies allocating need-oriented resources geared toward improving equality of continuous care should be developed.

## Introduction

The HIV epidemic, especially in developing countries, is still a challenge worldwide. With remarkable success and scaling up of antiretroviral therapy (ART) coverage, life expectancy and survival times of people living with HIV have significantly increased [1, 2]. The healthcare systems are challenged to maintain a growing number of those people [3, 4] who demand not only access to ART but also practical, comprehensive and continuous care aligned with life-long ART [5]. Regular monitoring of responses to HIV treatment, including monitoring of long-term drug toxicity, is recommended by current international guidelines [6–8]. Additionally, people living with HIV are more likely to suffer mental health disorders than the general population [9]. Routine assessment and management for mental health conditions, especially depression, are recommended to be integrated into the package of HIV care services for this group in non-psychiatric healthcare settings [7, 10].

Although there exists a vital role of ART and HIV continuous care, inequalities in accessing and utilizing HIV care are on a large scale. For example, previous studies reported a high rate of late presentation for HIV care in some Asian countries [11], a low rate of retention in care after the first initiation of ART in the United States [12], and low uptake of regular monitoring of virological and immunological outcomes as well as monitoring the toxicity of ART in resource-limited settings [13–15]. A systematic review revealed that only a few studies reported the rate of receiving routine ART toxicity monitoring at scheduled visits in follow-up settings [16]. Overall, contemporary data on access to and utilization of comprehensive HIV care after receiving ART, especially monitoring ART response and client-perceived uptake to HIV mental health care, are sparse.

As one of the developing countries in the Asia-Pacific region, China is also facing challenges in healthcare and managing a growing number of people living with HIV, of which there were 1.2 million in 2018 [17, 18]. In particular, China is one of only a few low- and middle-income countries in which the government is the major source of spending for care and treatment of people living with HIV (98.4%) [19]. China started the National Free Antiretroviral Treatment Program (NFATP) with national standardized follow-up medical records in 2003 and since then has expanded ART coverage to meet the WHO recommendation of "Treat-all" [3]. Under this program, China's free-ART manual was developed, in which routine blood tests accompanied by follow-up visits four times a year and at least one viral load test and CD4 count per year are free of charge for people who are stable on ART [20].

However, one free viral load test per annum is not enough. According to the Chinese guidelines, people who are stable on ART should have their viral load tested every six months to assess virological outcomes and identify virological failure to prevent drug resistance [21, 22]. Furthermore, CD4 remains the gold standard for identifying individuals with advanced disease [23]. A free CD4 test once a year is thus not adequate for people whose CD4 count is less than 300 cells/mm$^3$ [20, 21].

A systematic review showed that the median prevalence of symptoms of depression was greater than 60% among people living with HIV in China and highlighted the need for mental

healthcare for people living with HIV [24]. The Chinese national guidelines recommend that mental healthcare should be integrated into ART services when conditions permit [20, 21]. Still, there are no specific operating guidelines and rare reports of routine mental healthcare in Chinese ART cohorts.

Previous studies have indicated that socioeconomic status (SES), younger or older age, employment, education, area of residence, and health insurance were associated with HIV epidemics, HIV testing, maternal care, delays in HIV treatment, access to ART, attrition of ART, and even poorer treatment outcomes [11, 14, 25–28]. However, there is a paucity of studies measuring the socioeconomic-related inequality in care for people living with HIV who are stable on ART, especially in the Asia-Pacific region. Several Chinese studies suggested that there were disparities in HIV testing, access to HIV care or attrition of care, and viral suppression among women and key populations such as men who have sex with men [26, 29]. Nevertheless, those disparity measures did not consider socioeconomic and demographic factors, which may be social determinants that can influence health or healthcare as a whole [30]. Moreover, under the context of free ART to "Treat all" in China, no study has reported whether people living with HIV who have received ART are retained in the HIV care continuum and receive an equal amount of comprehensive care to maintain both their physical and mental health regardless of their socioeconomic status. Therefore, it is crucial to quantify the relationship between SES and comprehensive HIV care for people living with HIV who have received ART.

The relative concentration index and its decomposition have been widely used in studying socioeconomic-related inequalities in healthcare services [31–33]. Based on Wagstaff's suggestion, the decomposition of the concentration index can explain the source of healthcare inequality by the contribution of determinants [34]. Possible determinants can be categorized into "need" and "non-need" factors. Need factors are sources of variation in healthcare that are not considered to be "unfair". They are related to an individual's characteristics that are direct causes of medical services, usually, age and sex [35]. Non-need factors are sources of variation in health care that are "unfair" or "unjust", such as SES, education, and ethnicity [36].

The objectives of this study are two-fold. First, to determine whether there are socioeconomic-related inequalities in the uptake of comprehensive continuous care for people living with HIV in terms of maintaining physical and mental health, including services for retention in care, routine monitoring of ART toxicities, monitoring treatment outcomes, and routine assessment of mental health. Second, to decompose these inequalities by evaluating the contribution of socio-demographic factors.

## Materials and methods

### Study setting and participants

A hospital-based cross-sectional survey was conducted from April-November 2019 among people living with HIV attending a leading HIV treatment center in Kunming, the capital city of Yunnan province, southwestern China. Yunnan province has the highest number of PLWH in China [17]. At the end of 2018, there were about 15,000 people living with HIV in Kunming [37].

Individuals were eligible to be included in the study if they: 1) were aged 18 years or older, 2) had initiated ART and were currently receiving ART for more than one year but not more than five years, 3) were able to communicate in Chinese, and 4) were able to present themselves on the day of the interview and were sufficiently physically and mentally stable to provide verbal consent to participate in the study. The criterion for including only those on ART between 12 months and five years was set to ensure that the individuals had reached a

requirement of 12 months as the average time of retention in care for viral suppression as detailed elsewhere [12]. Furthermore, this criterion also can avoid some of the longer-term complications after five years, which need more intensive care, such as first-line regimen failure and drug resistance, resulting in a switch to a second-line regimen, with intensive viral load, CD4 count and medication adherence monitoring [38, 39].

All eligible people living with HIV who visited the HIV treatment center during the study period were invited to participate in this study.

## Sample size

The sample size was calculated using the finite population formula [40]. As no previous data on comprehensive care access among people living with HIV was available in China, it was conservatively estimated that comprehensive care access was 50%. There were nearly 8,000 people receiving ART in the HIV treatment center. Considering a margin of error of 4%, a confidence interval of 95%, and a 20% refusal rate or missing data, the minimum required sample size for the study was 664.

## Data collection

A structured questionnaire was developed in English and then translated into Chinese. Back-translation into English was also performed to validate the translation. Twenty persons living with HIV were interviewed in a pilot study to test the comprehensibility of the questionnaire. Five medical students were employed as research assistants and received a two-day training on the study protocol, interviewing skills, and data confidentiality by a field supervisor. A staff member of the HIV treatment center invited eligible participants into the study. The research assistant informed the study goals, benefits and risks of participation, and research procedures. The participants were told that they would be asked to anonymously respond to a structured questionnaire, which included socio-demographic characteristics. They were also informed that they could withdraw at any time and all information would be kept confidential. To ensure participant confidentiality, only verbal consent to participate in the study, including both questionnaire interviews and medical record reviews, was obtained. Each consenting participant provided their NFATP unique personal identification number in order to link their questionnaire data to their medical records.

After completing the interview, the field supervisor reviewed the participants' medical records in the database of NFATP to collect information on the number of HIV clinic visits in the last six months. The number of routine tests for toxicity monitoring, CD4 test, and viral load assay in the previous 12 months were also obtained.

## Variables

**Dependent variables.** The outcome variable of this study is the uptake of HIV comprehensive care. Based on Chinese and international guidelines [6–8, 21], comprehensive care in this study includes five binary variables, representing retention in care, treatment response monitoring, and mental healthcare, namely:

1. Retention in care was defined as having at least two HIV clinic visits in the last six months. Based on the NFATP free-ART manual, after initiating ART for more than one year, subsequent follow-up visits are scheduled every three months [20]. The dispensing of free-ART is three-monthly in the HIV treatment center to ensure enough antiretrovirals (ARVs) stocks for both newly initiated and continuously treated people. Therefore, retention in care in this study means that people living with HIV had adequate follow-up visits and

ARVs in the last six months. This variable was collected from the medical records of NFATP and the outpatient system of the HIV treatment center.

2. Routine monitoring of ART toxicities includes tests for renal function, liver function, and complete blood cell count performed at least four times per year. This indicator was based on a technical brief of the WHO on surveillance of antiretroviral toxicity [41] and the NFATP manual, which provides four free routine tests per year [20]. If a participant has at least four test records in the outpatient system, it means that the participant has received adequate ART toxicities monitoring.

3. Monitoring of immunological outcomes was defined as having adequate CD4 tests depending on the test results: a) for counts between 300 and 500 cells/mm$^3$, receiving at least one test in the last 12 months, and b) for counts less than 200 cells/mm$^3$, receiving at least two tests in the last 12 months [7, 21]. The NFATP provides free CD4 testing only once a year, and the extra tests must be paid for by the participants themselves or their health insurance, which are voluntary based on the clients' ability to pay. CD4 testing records were collected from the outpatient system and database of NFATP.

4. Monitoring of virologic outcomes was defined as having a plasma HIV viral load assay at least twice in the last 12 months. As with CD4 testing, the NFATP only provides free viral load testing once a year [20], so the additional viral load test was voluntary based on the client's ability to pay, and test costs can be beyond the client's affordability. Viral load testing records were collected from the outpatient system and NFATP database.

5. Self-reported mental health assessment was measured as perceived access to mental healthcare by the question "Have you received a mental health assessment from a health provider in the last 12 months?". Because there are no specific operating guidelines or records in the outpatient system or database of NFATP, we employed this indicator as perceived mental healthcare uptake. During the clinic visit, the mental health state can be assessed by medical staff free of charge.

**Socio-demographic characteristics.** Several socio-demographic factors associated with disparities in access to care may also be associated with inequalities in the HIV care continuum [11, 14, 16, 25–28, 30]. We investigated biological sex, age (at the time of the survey), ethnicity (Han and other), religious beliefs, marital status, education level (primary school or below, secondary school, high school, and university or above), employment status (employed and unemployed), and type of medical insurance (none, New Rural Cooperative Medical Insurance (NRCMI), Urban Employees Basic Medical Insurance (UEBMI), and Urban Residents Basic Medical Insurance (URBMI)). Possible determinants of healthcare uptake can be categorized into "need" and "non-need" factors. As mentioned earlier, "need" factors include age and sex [35] and other socio-demographic factors are "non-need" factors.

**Measuring socioeconomic status.** The wealth index per equivalent adult used to measure individual socioeconomic status was generated from self-reported household assets and housing conditions using principal component analyses, including ownership of a house, house size, number of bedrooms, type of floor, availability of drinking water, presence of a toilet, cooking fuel source, ownership of private vehicles, household furniture (i.e., table, chair, sofa, bed, wardrobe, and cupboard) and household appliances (i.e., television, refrigerator, washing machine, computer, microwave, mobile phone, and Internet broadband) [42]. Considering the household's size and demographic composition, we adjusted the wealth index by the number of equivalent adults in the household. The adult equivalents, *AE*, in the household is

derived from Eq (1) below [35]:

$$AE = (A + 0.5K)^{\theta} \tag{1}$$

where $A$ represents the number of adults in the household, $K$ refers to the number of children aged 14 years and younger, and $\theta$ is the degree of economies scale [43]. We used a weight of 1.0 for adults and 0.5 for those aged 14 years and younger, and the proposed value of 0.75 for $\theta$ to represent the degree of economies scale based on an empirical study in China [44].

For decomposition analysis, the wealth index was categorized by tercile (low, middle, and high).

**Measuring socioeconomic related inequality.** This study used a concentration index ($CI_x$) to measure socioeconomic-related inequality. It can be computed conveniently using the covariance between a healthcare outcome variable and the fractional rank in the socioeconomic position as Eq (2) [31]:

$$CI_x = \frac{2}{\mu} cov(y_i, R_i) \tag{2}$$

where $y_i$, $\mu$, and $R_i$ are the outcome variables of the $i^{th}$ individual, the mean or proportion of the outcome variable, and the fractional rank of the $i^{th}$ individual in the socioeconomic distribution, respectively.

The index typically ranges from -1 to +1, where a positive value emerges when uptake of healthcare is concentrated among the groups with higher SES, and a negative value means people in the lower SES group are more likely to receive healthcare than the rich. In practice, a value between 0.2 and 0.3 is regarded as a high degree of inequality [35]. Because of bounds dependence on the mean of the binary outcome, a normalized index, $CI_n$ was proposed by Wagstaff to correctly solve quantifying the degree of inequality within -1 to +1 [45] and can be written as follows:

$$CI_n = \frac{CI_x}{1 - \mu} \tag{3}$$

where $\mu$ refers to the mean of the outcome variable and $CI_x$ is the unnormalized concentration index.

**Decomposition of the concentration index.** To quantify the contribution of socio-demographic factors to observed healthcare inequalities of people living with HIV in Kunming, decomposition of the normalized concentration index ($CI_n$) was employed. We only focused on how socio-demographic explanatory variables influenced healthcare inequalities, so we did not include other health variables and clinical factors.

The decomposition method was first introduced for a linear, additive model for continuous outcome variables by Wagstaff *et al.* [34] as:

$$y = \alpha + \Sigma_k \beta_k x_k + \varepsilon \tag{4}$$

where $\beta_k$ is the coefficient of the explanatory variable $x_k$, and $\varepsilon$ is an error term.

The concentration index can be decomposed to the contribution to concentration index ($CC$), in which each contribution equals the product of the elasticity of socio-demographic factor to $y$ ($\beta_k \bar{x}_k / \mu$) and concentration index of $x_k$ ($C_k$), i.e. ($\beta_k \bar{x}_k / \mu$) $C_k$ [34], so the concentration index can be formulated as:

$$CI_x = \Sigma_k (\beta_k \bar{x}_k / \mu) C_k + GC_\varepsilon / \mu \tag{5}$$

where $\mu$ is the mean or proportion of $y$, $\bar{x}_k$ is the mean of $x_k$ and $GC_\varepsilon$ is the generalized

concentration index for the error term ($\varepsilon$). Eq (5) demonstrates that $CI_x$ is equal to a weighted sum of $k$ explanatory factors' concentration indices, i.e., $C_k$. $C_k$ reflects the distribution of SES by $k$ explanatory factors. For example, a positive $C_k$ of males means that males are concentrated among the rich. The residual term ($GC_\varepsilon / \mu$) implies the inequality in healthcare that is not explained by the systematic variation of explanatory factors, and it should be close to zero for a well-specified model [35].

However, because health sector outcome variables are intrinsically nonlinear, the decomposition approach is possible only if some linear approximation to the nonlinear model is performed. One common choice yielding probabilities in the range (0, 1) is the probit model, which is the standard additively normally distributed model. For binary outcome variables, one possibility is to use estimates of the marginal effects evaluated at the means [36]. That is, a nonlinear approximation for a binary outcome $y$ to Eq (6) with need variables and non-need variables is given by:

$$y_i = \alpha_j^m + \Sigma_j \beta_j^m x_{ji} + \Sigma_k \gamma_k^m z_{k_i} + \varepsilon_i^m \tag{6}$$

where $x$ and $z$ refer to need variables and non-need variables, and; $i$ refers to an individual; the $\beta_j{}^m$ and $\gamma_k{}^m$ are the marginal effects from the probit model of each variable treated as fixed parameters evaluated at the means, and $\varepsilon_i{}^m$ is the error term. Decomposition of the concentration index of a binary outcome based on the additive approximation regression of Eq (6) can be used [34], such that the normalized concentration index for $y_i$ can be yielded as:

$$CI_n = \Sigma_j (\beta_j^m \bar{x}_j / \mu) C_{jn} + \Sigma_k (r_k^m \bar{z}_k / \mu) C_{kn} + GC_\varepsilon / \mu \tag{7}$$

where the first term refers to the partial contribution of need variables, the second term is the contribution of non-need variables and SES, and the $C_{jn}$ and $C_{kn}$ are normalized concentration indices of need and non-need variables, respectively. A positive contribution percentage to the concentration index suggests that the combined marginal effect of explanatory factors and its distribution with respect to SES increases the size of inequality. If an explanatory factor makes a negative contribution percentage to the concentration index, the level of the pro-rich inequality in healthcare would be higher should that explanatory factor be removed.

To assess inequity in healthcare distribution, standardization of concentration index for differences in need variables is also important. The indirect standardization approach has demonstrated that one simply needs to deduct the contributions of the standardizing variables (including in the regression along with others) from the total concentration index, the index of horizontal inequity (i.e., indirectly standardized concentration index), obtained by deducting the contributions of need variables in Eq (7). The indirect standardized concentration index ($CI_x{}^{IS}$) can be explained as follows [35]:

$$CI_x{}^{IS} = CI_n - \Sigma_j (\beta_j^m \bar{x}_j / \mu) C_{jn} \tag{8}$$

## Statistical analysis

Data were entered using Epidata 3.1 and analyzed using R software version 4.0.1 and STATA/MP version 14.2 (Stata Corp. Lp, College Station, USA). Categorical variables were described as frequencies and percentages, while continuous variables were described as means and standard deviation (SD) or median value with interquartile range (IQR). The probabilities of dependent variables in different age groups, education levels, and SES groups were calculated and compared using the chi-square test for trend and Bonferroni's adjustment for multiple comparisons. We used a user-written Stata command "conindex" [46], which enables users to estimate the

Wagstaff normalized concentration index and p-value for testing that the index is equal to zero. We employed the "probit" model with all socio-demographic explanatory variables to obtain the marginal effect for the calculation of contributions to the concentration index.

## Ethical approval

The protocol of this study was approved by the Human Research Ethics Committee of Prince of Songkla University, Hat Yai, Songkhla, Thailand (REC: 61-340-18-1) and the Medical Ethics Committee, the Third People's Hospital of Kunming, Yunnan, China (REC: 2019012901). All researchers and data collectors were re-trained on the ethical issues prior to data collection. Before data collection, the aims of the study were presented to all participants. Confidentiality and anonymity of eligible participants were assured. Verbal consent was obtained from all subjects before the interviews. Participation in the study was voluntary, and participants could refuse to respond to any questions or discontinue their participation at any time. Unique codes were used to maintain the participants' confidentiality, and no personal identifiers were recorded. Because we conducted the study when participants were attending the routine visits, there was no compensation given to them.

## Results

### Socio-demographic characteristics of the respondents

Table 1 presents respondents' socio-demographic characteristics with their corresponding normalized concentration indices ($C_k$). A total of 702 adults living with HIV in Kunming participated in the study. The participants ranged in age from 18 to 77 years, with a median age of 41.0 years (IQR, 32.0 to 48.0 years). Most were under the age of 50 years (80.9%), male (69.4%), Han ethnicity (83.9%), and non-religious (83.9%). The percentages of married and single participants were 44.3% and 35.3%, respectively. About half were employed, over one-third achieved a secondary school level of education, and 22.7% had completed university. Basic medical insurance covered 92.7% of participants, including 44.3% using rural medical insurance with NRCMI and 30.5% urban residents with URBMI. Based on this medical insurance type, there was an approximately equal distribution of study participants in urban (48.3%) and rural (44.3%) sites.

There were significantly positive concentration indices among persons aged 18 to 34 years, married, with university or higher education level, and NRCMI insurance indicating that these factors were intensely concentrated among the rich. Conversely, participants who were aged more than 50 years, single, had achieved only secondary school education, and possessed no medical insurance were significantly more concentrated among the socioeconomically disadvantaged group.

### Uptake of HIV continuous care

Table 2 illustrates the distribution of complete uptake of HIV continuous care by socio-demographic characteristics. More than 80% of respondents reached the targets of retention in care in the past six months, monitoring of toxicities, and monitoring of immunological outcomes in the past 12 months, regardless of their SES. However, the proportions of those receiving adequate plasma viral load assay (15.8%) and mental health assessment services (15.0%) were low in the past 12 months.

The tendency of respondents to receive adequate immunological and virological monitoring significantly decreased with increasing age ($p < 0.001$). Single subjects accounted for the highest proportion (20.6%) of receiving adequate monitor of virologic outcomes ($p = 0.047$).

**Table 1. Distribution and concentration indices of socio-demographic characteristics.**

|  | N (%) | $C_k$ | p- value |
|---|---|---|---|
| Total | 702 (100.0) |  |  |
| **Biological sex** |  |  |  |
| Male | 487 (69.4) | 0.0744 | 0.116 |
| Female | 215 (30.6) | -0.0744 | 0.116 |
| **Age (years)** |  |  |  |
| 18–34 | 244 (34.8) | 0.1070 | 0.019 |
| 35–49 | 324 (46.2) | -0.0163 | 0.710 |
| ≥50 | 134 (19.1) | -0.1316 | 0.018 |
| **Ethnicity** |  |  |  |
| Han | 589 (83.9)) | -0.0138 | 0.817 |
| Other | 113 (16.1) | 0.0138 | 0.817 |
| **Religious belief** |  |  |  |
| Yes | 589 (83.9) | 0.0673 | 0.257 |
| No | 113 (16.1) | -0.0673 | 0.257 |
| **Marital status** |  |  |  |
| Married | 311 (44.3) | 0.1949 | <0.001 |
| Single | 248 (35.3) | -0.2660 | <0.001 |
| Divorced | 110 (15.7) | -0.0454 | 0.320 |
| Widowed | 33 (4.7) | -0.0572 | 0.579 |
| **Education level** |  |  |  |
| < = Primary school | 174 (24.8) | -0.0657 | 0.194 |
| Secondary school | 245 (34.9) | -0.0901 | 0.049 |
| High school | 124 (17.7) | 0.0137 | 0.811 |
| > = University | 159 (22.7) | 0.1754 | 0.001 |
| **Employed status** |  |  |  |
| Employed | 387 (55.1) | 0.0217 | 0.621 |
| Unemployed | 315 (44.9) | -0.0217 | 0.621 |
| **Medical insurance** |  |  |  |
| None | 51 (7.3) | -0.3401 | <0.001 |
| NRCMI | 311 (44.3) | 0.0994 | 0.024 |
| UEBMI | 126 (18.0) | 0.1011 | 0.075 |
| URBMI | 214 (30.5) | -0.0778 | 0.101 |

SES, socioeconomic status; $C_k$, concentration index of socio-demographic factors; NRCMI, New Rural Cooperative Medical Insurance; UEBMI, Urban Employees Basic Medical Insurance; URBMI, Urban Residents Basic Medical Insurance. P-value is for concentration index.

Employed participants obtained more frequently adequate virological monitoring than the unemployed ($p$ <0.001). There was also a significant difference in the proportions of those receiving monitoring for virological outcomes by type of medical insurance ($p$ = 0.034).

Respondents in the low SES group reported the lowest percentage of uptake of mental health assessment (10.3%) followed by those in the high (16.7%) and middle SES groups (17.9%), and these differences were statistically significant ($p$ = 0.044). A significant sex disparity was also found for the proportions of those receiving mental health assessment ($p$ = 0.036). The proportion of people undergoing mental health assessments increased significantly with increasing education level ($p$ = 0.004). Participants with religious beliefs reported significantly higher rates of receiving mental health assessments than those without religious beliefs ($p$ = 0.004).

**Table 2. Distribution of complete uptake of HIV continuous care by socio-demographic characteristics.**

| Variable | RIC (%) | RMT (%) | MIO (%) | MVO (%) | AMH (%) |
|---|---|---|---|---|---|
| **Total** | 82.9 | 81.9 | 80.6 | 15.8 | 15.0 |
| **SES group** | $p = 0.713$ | $p = 0.904$ | $p = 0.349$ | $p = 0.899$ | $p = 0.044$ |
| Low | 80.8 | 81.2 | 77.4 | 15.0 | 10.3 |
| Middle | 86.8 | 82.1 | 82.5 | 17.5 | 17.9 |
| High | 81.3 | 82.5 | 82.1 | 15.0 | 16.7 |
| **Biological sex** | $p = 0.957$ | $p = 0.194$ | $p = 0.242$ | $p = 0.116$ | $p = 0.036$ |
| Male | 83.0 | 83.2 | 79.5 | 17.2 | 16.8 |
| Female | 82.8 | 79.1 | 83.3 | 12.6 | 10.7 |
| **Age (years)** | $p = 0.557$ | $p = 0.167$ | $p < 0.001$ | $p < 0.001$ | $p = 0.200$ |
| 18–34 | 81.1 | 81.6 | 88.5 | 23.4 | 19.3 |
| 35–49 | 84.3 | 79.3 | 78.4 | 13.0 | 13.3 |
| 50+ | 82.8 | 88.8 | 71.6 | 9.0 | 11.2 |
| **Ethnicity** | $p = 0.527$ | $p = 0.343$ | $p = 0.817$ | $p = 0.084$ | $p = 0.600$ |
| Han | 82.5 | 82.5 | 80.5 | 14.8 | 15.3 |
| Minority | 85.0 | 78.8 | 81.4 | 21.2 | 13.3 |
| **Religions belief** | $p = 0.147$ | $p = 0.515$ | $p = 0.817$ | $p = 0.420$ | $p = 0.004$ |
| Yes | 87.6 | 84.1 | 81.4 | 13.3 | 23.9 |
| No | 82.0 | 81.5 | 80.5 | 16.3 | 13.2 |
| **Marital status** | $p = 0.252$ | $p = 0.962$ | $p = 0.119$ | $p = 0.047$ | $p = 0.100$ |
| Married | 81.4 | 81.4 | 78.1 | 13.5 | 12.5 |
| Single | 86.7 | 81.9 | 85.5 | 20.6 | 18.2 |
| Divorced | 80.0 | 83.6 | 77.3 | 14.6 | 17.7 |
| Widowed | 78.8 | 81.8 | 78.8 | 6.1 | 6.1 |
| **Education level** | $p = 0.145$ | $p = 0.500$ | $p = 0.081$ | $p = 0.062$ | $p = 0.004$ |
| ≤Primary school | 81.6 | 82.2 | 83.9 | 13.8 | 10.9 |
| Secondary school | 80.8 | 79.6 | 73.9 | 13.9 | 13.1 |
| High school | 83.9 | 83.9 | 77.4 | 16.1 | 15.3 |
| ≥University | 86.8 | 83.7 | 89.9 | 20.8 | 22.0 |
| **Employment status** | $p = 0.710$ | $p = 0.429$ | $p = 0.085$ | $p < 0.001$ | $p = 1.000$ |
| Employed | 82.4 | 82.9 | 82.9 | 20.7 | 15.0 |
| Unemployed | 83.5 | 80.6 | 77.8 | 9.8 | 14.9 |
| **Medical insurance** | $p = 0.128$ | $p = 0.270$ | $p = 0.117$ | $p = 0.034$ | $p = 0.060$ |
| None | 86.3 | 74.5 | 82.4 | 19.6 | 19.6 |
| NRCMI | 79.4 | 83.0 | 80.4 | 16.7 | 10.9 |
| UEBMI | 88.1 | 85.7 | 87.3 | 21.4 | 16.7 |
| URBMI | 84.1 | 79.9 | 76.6 | 10.3 | 18.7 |

RIC, retention in care; RMT, routine monitoring of toxicities; MIO, Monitoring of immunological outcome; MVO, Monitoring of virologic outcome; AMH, Self-reported assessment of mental health; NRCMI, New Rural Cooperative Medical Insurance; UEBMI, Urban Employees Basic Medical Insurance; URBMI, Urban Residents Basic Medical Insurance.

## Concentration indices of HIV continuous care outcomes and contribution of socio-demographic factors

Results of concentration indices of dependent variables and their aggregated contribution of socio-demographic factors are displayed in Table 3. A statistically significant concentration index for the uptake of mental health assessment was found ($CI_n = 0.1337$, $p = 0.029$), indicating a pro-rich inequality. After controlling for unequal need distribution (age and sex), the

**Table 3. Concentration index and aggregated contribution of regressors to concentration indices for HIV care continuum.**

|  | RIC | RMT | MIO | MVO | AMH |
|---|---|---|---|---|---|
|  | AC (CC%) | AC (CC%) | AC (CC%) | AC (CC%) | AC (CC%) |
| $CI_n$ | 0.0117 | 0.0315 | 0.0736 | -0.0308 | 0.1337* |
| $CI_x^{IS}$ | 0.0157 | 0.0344 | 0.0707 | -0.0501 | 0.1123* |
| **Need factors** |  |  |  |  |  |
| Age | -0.0031 (-26.9) | -0.0037 (-11.9) | 0.0051 (7.0) | 0.0171 (-55.5) | 0.0149 (11.1) |
| Biological sex | -0.0009 (-7.5) | 0.0008 (2.7) | 0.0240 (-3.0) | 0.0022 (-7.0) | 0.0065 (4.9) |
| Subtotal | **-0.0040 (-34.3)** | **-0.0029 (-9.2)** | **0.0291 (4.0)** | **0.0193 (-62.5)** | **0.0214 (16.0)** |
| **Non-need factors** |  |  |  |  |  |
| SES | 0.0046 (39.6) | 0.0018 (5.8) | 0.0117 (15.9) | -0.0008 (2.7) | 0.1670 (124.9) |
| Ethnicity | 0.0000 (0.3) | -0.0001 (-0.3) | -0.0001 (-0.1) | 0.0006 (-2.1) | -0.0004 (-0.3) |
| Religious belief | 0.0044 (37.3) | 0.0017 (5.5) | 0.0005 (0.7) | -0.0101 (32.7) | 0.0365 (27.3) |
| Marriage status | -0.0009 (-8.3) | -0.0018 (-5.5) | -0.0009 (0.0) | -0.0074 (24.1) | -0.0157 (-11.8) |
| Education level | 0.0017 (14.1) | 0.0020 (6.3) | 0.0036 (4.9) | -0.0089 (22.2) | 0.0108 (8.1) |
| Employment status | -0.0002 (-1.4) | 0.0003 (0.9) | 0.0000 (0.0) | 0.0046 (-15.0) | -0.0016 (-1.2) |
| Medical insurance | -0.0015 (-13.0) | 0.0049 (15.6) | 0.0002 (0.2) | 0.0004 (-1.5) | -0.0319 (-23.9) |
| Subtotal | **0.0037 (68.7)** | **0.0079 (28.3)** | **0.0150 (21.6)** | **-0.0216 (63.4)** | **0.1647 (123.0)** |
| **Residual (unexplained)** | -0.0040 | 0.0256 | 0.0557 | -0.0306 | -0.0524 |

$CI_n$, normalized concentration index; $CI_x^{IS}$, Indirectly standardized concentration index; RIC, retention in care; RMT, routine monitoring of toxicities; MIO, Monitoring immunological outcome; MVO, Monitoring virological outcome; AMH, Self-reported assessment of mental health; AC, absolute contribution to concentration index; CC%, percentage of contribution to concentration index;

*p-value <0.05.

indirect standardized concentration index remained significant for uptake of mental health assessment ($CI_x^{IS}$ = 0.1127, $p$ = 0.042).

Non-significant pro-rich inequalities were found for retention in care ($CI_n$ = 0.1168, $p$ = 0.840), routine monitoring of toxicities ($CI_n$ = 0.0315, $p$ = 0.579), and monitoring of immunological outcomes ($CI_n$ = 0.0736, $p$ = 0.182). However, the opposite direction was found for the uptake of adequate virological monitoring with a concentration index of -0.0308 ($p$-value 0.607), indicating pro-poor inequality slightly. Subtracting the contribution of need variables (i.e., age and sex), the need-adjusted concentration indices for retention in care, routine monitoring of toxicities, and monitoring immunological outcome were 0.0157, 0.0344, and 0.0707, respectively, showing the same direction of pro-rich horizontal inequality. The need-adjusted concentration index of monitoring virological outcomes was -0.0501, indicating a stronger horizontal inequality degree favoring the poor.

Table 3 also shows the contribution of socio-demographic characteristics on socioeconomic-related inequalities where a positive contribution percentage increases inequality, and a negative contribution percentage decreases inequality. The utilization of mental health assessment, a significant pro-rich inequality, is taken as an example to illustrate the decomposition of a concentration index into its determinants. Participants' SES (124.9%), religious belief (27.3%), and age (11.1%) had the highest positive contributions to the measured inequality. In contrast, participants' medical insurance (-23.9%) and marital status (-11.8%) had negative contributions to the pro-rich inequality of mental health assessment, that is, these factors decreased the inequality size in the utilization of mental health assessment. The residuals of the regression models (-0.0524, -39.2%) implied a large unexplained proportion of factors contributing to the concentration index of mental health assessment. SES positively contributed to

inequalities of all dependent variables, with the highest contribution for assessing mental health (124.9%) and the lowest for monitoring virological outcomes (2.7%).

For the non-significant concentration index, these factors' positive and negative contributions were canceled out. Need variables (age and sex) provided the main negative aggregated contribution to concentration indices of adequate monitoring of virological outcomes (-62.5%), retention in care (-34.3%), and routine monitoring of ART toxicities (-9.2%), but mild positive contribution for monitoring immunological outcomes (4.0%). On the other hand, having a religious belief provided high contributions to inequalities for retention in care (37.3%) and virological monitoring (32.7%). Medical insurance offered positive contributions for routine monitoring of toxicities and monitoring immunological outcomes and negative contributions to the other outcome variables. Ethnicity and employment status had tiny contributions to inequalities of all outcomes.

## Discussion

### Main findings and comparison with previous studies

This study expands other previous research by applying a concentration index to examine the presence of socioeconomic-related inequality in comprehensive continuous care for people living with HIV who had been on ART more than one year but less than five years the context of "national free ART" in Kunming, the epicenter of HIV in China. We also identified the contribution of some socio-demographic characteristics to the inequality by a decomposition approach.

In the present study, for the first time, the socioeconomic-related inequality in receiving mental healthcare in China was measured and explained. We found a low rate of having mental health assessment (15%), which was consistent with other studies around the world [47]. A systematic review reported that people living with HIV were at risk for mental health problems in their lifetimes, and this is true in China as this vulnerable group suffered more severe discrimination and lack of available resources than did those living in other countries [24]. Failure to be screened and treated for these psychological disorders may hamper the successful treatment of their HIV infection [10]. To integrate mental healthcare and ART services, a training course for health providers, an action guideline, and an information system with a standardized procedure for assessing and recording mental status are needed [48].

We also found that perceived uptake to mental health assessment was disproportionately concentrated among people with a higher socioeconomic status. This is consistent with inequalities in the utilization of specialty mental health services among persons living with HIV in the USA [49]. These results agree with the general global consensus around the relationship between socioeconomic inequality and pro-rich healthcare utilization under the shrinkage of financial subsidy and fund support [50]. Moreover, we found that the main positive contributor to inequality of mental health assessment was SES, which accounted for 124.9%, indicating that a higher socioeconomic position may increase the size of the pro-rich inequality compared to those in a lower socioeconomic group. Apart from receiving free mental health assessment in the HIV treatment center, the rich may also receive mental health assessment in specialized psychiatric settings, which is not free. This explains how SES plays a role in a pro-rich inequality of this type of care.

Other socioeconomic and socio-demographic factors also play a certain role in the inequality of receiving mental health assessments. Religious belief positively contributed to the pro-rich inequality of mental health assessment. Participants with religious beliefs were concentrated in the high SES group (positive $C_k$), and they tended to have more retention in care, so they increased their chance of receiving mental health assessment. Medical insurance accounted for a negative contribution to pro-rich on mental health assessment, suggesting that

having medical insurance and the choice of medical insurance may affect the uptake of mental health assessment and health services utilization [51], as the cost of mental healthcare can be covered by medical insurance.

Our study indicated that under the national free ART program in China, inequalities of the utilization of all other comprehensive continuous care for people living with HIV who were stable and on ART in Kunming were not significant. Retention in care is a critical indicator of success in long-term HIV medication and the necessary component of a successful treatment-as-prevention strategy [12]. Our study showed that the majority of participants were retained in the care of the free ART program with clinic visit-based indicators in Kunming. This was consistent with recent studies in China and some other Asian countries [3, 11].

With the need for life-long ART and long-term toxicities of ART, our study found that more than 80% of participants received routine regular laboratory monitoring of toxicities of ART, which meant that NFATP in Kunming put surveillance of antiretroviral toxicity as a national priority indicator of the health sector response to HIV and agreed with the technical brief given by the WHO [41]. Toxicities appeared to be a relative reason for disengagement from care [3]. A study in Thailand demonstrated that simple and inexpensive monitoring of key biomarkers was feasible at some time points [15].

Different directions were shown for monitoring treatment outcomes in our study. Most participants received adequate immunological response monitoring, whereas 84.2% lacked adequate routine virological monitoring [21], which was lower than in a previous study in sub-setting in sub-Saharan Africa [52]. Interestingly, virological response monitoring was non-significant in favor of the poor in our research, and SES contributed minimally to the uptake of intensive viral load testing in Kunming. As an additional viral load test has to be paid out of pocket or by medical insurance based on the client's ability to pay, test costs can be beyond the client's affordability, neither the rich nor the poor might be willing to pay this fee. This explained the low uptake rate of adequate viral load test. The policies for HIV response, services, and core indicators in China for viral load testing should be adjusted practically to close the gap in viral load testing [53, 54].

Age provided a main negative aggregated contribution to reduce pro-rich inequalities of adequate virological monitoring (-62.5%), retention in care (-34.3%), and routine monitoring of toxicities (-9.2%). As the prolongation of life expectancy of people aging with HIV, this population is also facing an aging problem, similar to the general population. We found that age was the main contributor between the need variables, suggesting that need-oriented utilization of health care can reduce the degree of inequality to meet the needs of HIV care in Kunming.

## Strengths and limitations

This study used comprehensive indicators to cover physical and mental aspects of long-term care for people living with HIV on ART rather than a single specific indicator to measure healthcare utilization. To our knowledge, this is the first study to measure and explain the inequality of a series of comprehensive HIV continuous care by socio-demographic factors. We provide an informative picture of thorough care for the HIV care continuum among different socioeconomic groups in China.

There were several limitations in this study. First, the household assets and conditions were self-reported, which might result in both under- or over-reporting of participants' SES. Second, this study's findings are only valid for the population of Kunming city using hospital facilities. The inclusion of healthcare users outside the hospital setting would provide a clearer picture of the real inequality gap among the whole healthcare system for people living with HIV who had been on ART in Kunming. National multicenter studies on all-integrated

healthcare for people living with HIV would provide such information. Third, the lack of qualitative data, such as pro-rich inequality in mental health assessment and pro-poor inequality in virological monitoring, limited our ability to explain inequality in another dimension. The nature of a cross-sectional study limited its ability to make causal inferences. Finally, a sizeable unexplained proportion of contribution suggested that other mechanisms that we were unable to measure played a substantial role in inequality in the utilization of comprehensive continuous care. Further studies are needed to illuminate the impact of the scope from individual factors such as clinical characteristics or medication adherence, healthcare providers' role, the performance of the healthcare system, and other unmeasured variables.

### Implications from the study

Considering the low coverage of intensive virological monitoring and mental health assessment found in this study, there is an urgent need for (i) increasing the awareness of viral load monitoring and mental health assessment for people living with HIV among clinicians, (ii) improving the attention on the importance of performing viral load testing and mental health assessment among people living with HIV during their long-term treatment period, (iii) reliable return of results, point-of-care viral load testing in healthcare facilities, (iv) well-trained professional mental health service personnel and facilities for people living with HIV, and (v) reducing the cost of viral load testing. To diminish the degree of inequality of HIV care utilization under a free ART context, implications include (i) strengthening governmental policies, welfare, and social support to reduce the gap between the rich and poor, including vulnerable people living with HIV, (ii) response by the civil affairs department to improve the assistance system for people living with HIV, (iii) a lower user-pay amount from medical insurance companies and an expansion in the scope of medical insurance reimbursement to support access to ancillary long-term HIV care such as mental health services, (iv) the allocation of medical care resources based on the needs of people living with HIV which can reduce the degree of the socioeconomic-related inequality, and (v) addressing staffing and resource limitations around HIV comprehensive care.

### Conclusions

In Kunming City of China, there is a higher prevalence of retention in care, monitoring of toxicities, and immunological outcomes, but lower rates of completing adequate virological monitoring and self-reported mental health assessment receipt among people living with HIV receiving ART. We found that under the national free ART program in Kunming, pro-rich inequality of the utilization of mental health assessment was significant, but no significant inequalities of other comprehensive continuous care for people living with HIV were found in Kunming. We also found that socioeconomic status positively contributed to inequalities in mental health assessment. Between the two need variables (age and sex), age contributed more to the inequalities in the utilization of all HIV continuous care. This implies that the degree of such inequalities can be reduced should comprehensive care be provided proportionately to people living with HIV of all age groups. These findings can provide evidence for policymakers to develop policies that allocate need-oriented healthcare utilization geared toward more equality in comprehensive continuous care for people living with HIV.

### Supporting information

**S1 File. Study questionnaire (English and Chinese version).**
(DOCX)

**S2 File. Minimal dataset.**
(CSV)

**S1 Checklist. STROBE statement—Checklist of items that should be included in reports of observational studies.**
(DOCX)

## Acknowledgments

We highly appreciate all staff members of the HIV treatment center who support our study. We gratefully acknowledge Professor Alan Geater and Associated professor Edward McNeil from Prince of Songkla University, Songkhla, Thailand, for checking the R and Stata commands that helped this analysis.

## Author Contributions

**Conceptualization:** Yongmei Jin, Sawitri Assanangkornchai.

**Data curation:** Yongmei Jin, Yingrong Du, Jingsong Bai, Yongrui Yang.

**Formal analysis:** Yongmei Jin, Sawitri Assanangkornchai, Jun Liu.

**Funding acquisition:** Yongmei Jin.

**Investigation:** Yongmei Jin, Yingrong Du, Jun Liu, Jingsong Bai, Yongrui Yang.

**Methodology:** Yongmei Jin, Sawitri Assanangkornchai.

**Project administration:** Yongmei Jin, Sawitri Assanangkornchai, Yingrong Du.

**Resources:** Yingrong Du, Jun Liu, Jingsong Bai, Yongrui Yang.

**Supervision:** Sawitri Assanangkornchai.

**Validation:** Yongmei Jin, Sawitri Assanangkornchai.

**Writing – original draft:** Yongmei Jin.

**Writing – review & editing:** Sawitri Assanangkornchai.

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
