## [Decision Letter · Decision Letter 0]

9 Feb 2021

PONE-D-20-28608

Measuring and Explaining Inequality of Continuous Care for People Living With HIV/AIDS on Antiretroviral Therapy in Kunming , China

PLOS ONE

Dear Dr. Assanangkornchai,

Thank you for submitting your manuscript to PLOS ONE. After careful consideration, we feel that it has merit but does not fully meet PLOS ONE’s publication criteria as it currently stands. Therefore, we invite you to submit a revised version of the manuscript that addresses the points raised during the review process.

We look forward to receiving your revised manuscript.

Kind regards,

Siyan Yi, MD, MHSc, PhD

Academic Editor

PLOS ONE

Additional Editor Comments:

We apologize for the delay in receiving comments from reviewers. Only one reviewer submitted his/her comments and suggested rejection. However, I would like to give the authors another chance to revise and resubmit the paper.

Overall, the quality of writing left the reviewers in difficulty judging the study. Please follow standard scientific report writing rules (e.g., use of tenses and terminologies, spelling out numbers smaller than 10, use of tenses and punctuations). I have spotted several easy grammatical errors, typos, and complex sentences across the manuscript. Many statements also require supporting references and clarification. In the text, please minimize the use of uncommon and unnecessary abbreviations (IRC, RTM, MIO, MVO, AMH, etc.) that reduce the manuscript’s readability. The manuscript is not publishable in its current form and will benefit from thorough proofreading by a native or more experienced academic writer. Below are some suggestions, which may not be exhaustive. We will decide whether to continue reviewing the paper based on the improvement made in the next submission.

Title and keywords:

 Please revise the title according to PLOS’ styles and guidelines (e.g., do not start each word with a capital). Please remove ‘AIDS’ from ‘people living with HIV.’ UNAIDS recommended that ‘HIV/AIDS’ should not be used since 2015. Also, ‘people living with HIV’ should not be abbreviated or addressed as patients. This is applied across the paper, including keywords. See: https://www.unaids.org/sites/default/files/media_asset/2015_terminology_guidelines_en.pdf To increase the visibility of your paper, please avoid keywords that are overlapping with those in the title. This may help: https://falconediting.com/en/blog/6-tips-for-choosing-keywords-for-your-scientific-manuscript For PLOS journals, keywords should be included in the submission system, not in the text.

Abstract:

 A structured abstract is recommended (background, methods, results, conclusions). In Methods, please provide the study period and summarize the sampling methods. Results should provide the sample size and essential characteristics (age, sex).

Introduction:

 The Introduction contains long and complex sentences that are difficult to read and need to be re-structured.

Materials and Methods:

 Inclusion criteria should be revised to improve the quality of writing and understanding and avoid redundancy. For example, the first sentence under the ‘Study setting and participants’ stated that this study included people living with HIV who had initiated ART for at least one year, but not more than five years. This information was repeated in inclusion criteria 3, and it was not entirely consistent. Was sample size calculation performed? Please provide a reason for excluding people living with HIV aged 18 and those who had received ART more than five years. Inclusion criteria 3 is difficult to read, and criteria 5 is not understandable. What does ‘physical and psychological ability’ mean? What happened if someone could not complete the questionnaire in 30 minutes? Please provide references to support variable measurements. I am not quite sure if the criteria remained applicable in China in 2019. For example, are people living with HIV are still required to visit an ART clinic two times or more in 90 days? Is mental health a component in the treatment and care service package? Please be sensitive in using the term ‘gender’ and ‘sex.’ Please double-check if ‘gender’ is the correct term for this study context. Line 163: What does ‘need or non-need factor’ mean? Data analysis needs further details and should be aligned with the result presentation. For example, I am not sure if I understand why the authors compare socio-demographic characteristics in SES groups (Table 1), which is quite predictable. In the data analyses, the authors mentioned about the comparison of dependent variables in different groups. It was also not clear whether the analyses were controlled for confounding factors. If so, how were the models built, or how were the covariates selected?  Only Bonferroni's adjustment was mentioned. Ethics: Describe the measures taken to protect participants’ privacy and confidentiality. Also, did the participants receive any compensation?

Results

 The wording and terminologies used in the description of the results need improvement. Descriptive results can be summarized as they are mostly repeating the information in tables. Table 2: Exact p values should be presented rather than showing *p-value <0.05, ** p-value <0.01, and ***p-value <0.001 in the footnote.

Discussion

 Please refine the discussion according to the earlier comments and reviewer’s comments, focusing on this study's significant findings. It is also important to discuss the socio-economic burden of HIV care continuum on people living with HIV, particularly in the study setting context (e.g., free vs. charged services, other costs).

References

 The reference list needs to be revised to be aligned with PLOS’s styles and requirements.

Journal Requirements:

2. Please include additional information regarding the survey or questionnaire used in the study and ensure that you have provided sufficient details that others could replicate the analyses. For instance, if you developed a questionnaire as part of this study and it is not under a copyright more restrictive than CC-BY, please include a copy, in both the original language and English, as Supporting Information, or include a citation if it has been published previously.

Reviewers' comments:

Reviewer's Responses to Questions

**Comments to the Author**

1. Is the manuscript technically sound, and do the data support the conclusions?

Reviewer #1: Partly

2. Has the statistical analysis been performed appropriately and rigorously? 

Reviewer #1: Yes

3. Have the authors made all data underlying the findings in their manuscript fully available?

Reviewer #1: No

4. Is the manuscript presented in an intelligible fashion and written in standard English?

Reviewer #1: No

5. Review Comments to the Author

Reviewer #1: 1) Many concentration indices were with non-significant p-values, but the authors still interpreted as if there were effect, which is inappropriate. The results of the manuscript should be summarized in a succinct manner with the significant and important results stressed.

2) Table 3, it is unclear to me for the same variable why the absolute contribution and percentage of contribution to concentration index can have different signs?

3) Please ask for professional editing help with English language.

6. PLOS authors have the option to publish the peer review history of their article (what does this mean?). If published, this will include your full peer review and any attached files.

Reviewer #1: No

---

## [Author Response · Author response to Decision Letter 0]

18 Mar 2021

Responses to Editor and Reviewer

PLOS One Manuscript number: PONE-D-20-28608

Measuring and explaining inequality of continuous care for people living with

HIV on antiretroviral therapy in Kunming, China

Additional Editor Comments:

1.Overall, the quality of writing left the reviewers in difficulty judging the study. Please follow standard scientific report writing rules (e.g., use of tenses and terminologies, spelling out numbers smaller than 10, use of tenses and punctuations). I have spotted several easy grammatical errors, typos, and complex sentences across the manuscript. Many statements also require supporting references and clarification. In the text, please minimize the use of uncommon and unnecessary abbreviations (IRC, RTM, MIO, MVO, AMH, etc.) that reduce the manuscript’s readability. The manuscript is not publishable in its current form and will benefit from thorough proofreading by a native or more experienced academic writer. Below are some suggestions, which may not be exhaustive. We will decide whether to continue reviewing the paper based on the improvement made in the next submission.

Response: Thank you for your valuable comments. We have checked and made revisions accordingly. We have also asked a native English academic to edit the final manuscript.

2.Title and keywords:

• Please revise the title according to PLOS’ styles and guidelines (e.g., do not start each word with a capital).

Response: Checked and revised accordingly.

• Please remove ‘AIDS’ from ‘people living with HIV.’ UNAIDS recommended that ‘HIV/AIDS’ should not be used since 2015. Also, ‘people living with HIV’ should not be abbreviated or addressed as patients. This is applied across the paper, including keywords. See: https://www.unaids.org/sites/default/files/media_asset/2015_terminology_guidelines_en.pdf

Response: Thank you for your advice. We have checked and made revisions accordingly.

• To increase the visibility of your paper, please avoid keywords that are overlapping with those in the title. This may help: https://falconediting.com/en/blog/6-tips-for-choosing-keywords-for-your-scientific-manuscript

Response: Thank you for your suggestions. We have modified the keywords as: HIV, healthcare utilization, healthcare inequality, concentration index, decomposition, China.

• For PLOS journals, keywords should be included in the submission system, not in the text.

Response: We have made revisions accordingly.

3. Abstract:

• A structured abstract is recommended (background, methods, results, conclusions).

Response: We have made revisions accordingly.

• In Methods, please provide the study period and summarize the sampling methods.

Response: We have made revisions accordingly. 

• Results should provide the sample size and essential characteristics (age, sex).

Response: We have added the sample size and essential characteristics (age, sex) in the methods part.

4. Introduction:

• The Introduction contains long and complex sentences that are difficult to read and need to be restructured.

Response: We have rewritten the introduction part.

5. Materials and Methods:

• Inclusion criteria should be revised to improve the quality of writing and understanding and avoid redundancy. For example, the first sentence under the ‘Study setting and participants’ stated that this study included people living with HIV who had initiated ART for at least one year, but not more than five years. This information was repeated in inclusion criteria 3, and it was not entirely consistent.

Response: We have made revisions accordingly.

(Lines 152-156: “Individuals were eligible to be included in the study if they: 1) were aged 18 years or older; 2) had HIV infection clinically diagnosed and confirmed with serological tests; 3) had initiated ART and were currently receiving ART for more than 1 year but not more than 5 years; 4) were able and willing to provide verbal consent to participate in the study, and; 5) were able to communicate in Chinese.”)

• Was sample size calculation performed?

Response: Yes, we have added the sample size calculation in the revised manuscript. 

(Lines 164-170: “The sample size was calculated using the finite population formula. As no previous data on comprehensive care access among PLWH was available in China, it is conservatively estimated that comprehensive care access was 50%, there were nearly 8,000 people receiving ART in the HIV treatment center, considering a margin of error of 4%, a confidence interval of 95% and a 20% refusal rate or missing data. Finally, the minimum required sample size for the study was 664.”)

• Please provide a reason for excluding people living with HIV aged 18 and those who had received ART more than five years.

Response: We did not exclude people living with HIV aged 18 years. We apologize for the confusing sentence in the previous version. In this version, we have rewritten the inclusion criteria to make them clearer. (Lines 152-153) 

• Inclusion criteria 3 is difficult to read, and criteria 5 is not understandable. What does ‘physical and psychological ability’ mean? What happened if someone could not complete the questionnaire in 30 minutes?

Response: We have checked and made revisions accordingly. We have changed “physical and psychological ability” to “were able and willing to provide verbal consent to participate in the study and were able to communicate in Chinese”. (Lines 155-156)

• Please provide references to support variable measurements. I am not quite sure if the criteria remained applicable in China in 2019. For example, are people living with HIV are still required to visit an ART clinic two times or more in 90 days? Is mental health a component in the treatment and care service package?

Response: We have made revisions accordingly (Lines 196-224). People living with HIV were still required to visit the ART clinic every 3 months in 2019. Based on the free ART manual of China free antiretroviral treatment program (NFATP), after initiating ART for more than one-year, subsequent follow-up visits are scheduled every 3 months, and the dispensing of free ART is three-monthly in the HIV treatment center to ensure enough stocks of ART for both newly initiated and continuously treated people. Mental health is not a standard component in the ART service package. However, Chinese national guidelines recommend that mental healthcare can be integrated into ART services when conditions permit, but there are no specific operating guidelines or records in the outpatient system or database of NFATP.

• Please be sensitive in using the term ‘gender’ and ‘sex.’ Please double-check if ‘gender’ is the correct term for this study context.

Response: We have checked and made revisions accordingly. We have used “sex” or “biological sex” throughout the manuscript. 

• Line 163: What does ‘need or non-need factor’ mean?

Response: As stated in the Introduction (lines 131-135) and Materials and methods (Lines 233-235), “Possible determinants (of healthcare inequality) can be categorized into “need” and “non-need” factors. Need factors are sources of variation in healthcare that are not considered to be “unfair”. They are related to characteristics of an individual that are direct causes of the use of medical services, usually age and sex. Non-need factors are sources of variation in health care that are “unfair” or “unjust”, such as SES, education, and ethnicity.” In this study, “need” factors include age and sex and other socio-demographic factors are “non-need” factors.

• Data analysis needs further details and should be aligned with the result presentation. For example, I am not sure if I understand why the authors compare socio-demographic characteristics in SES groups (Table 1), which is quite predictable. In the data analyses, the authors mentioned about the comparison of dependent variables in different groups. It was also not clear whether the analyses were controlled for confounding factors. If so, how were the models built, or how were the covariates selected? Only Bonferroni's adjustment was mentioned.

Response: We have revised Table 1 to show the distribution and concentration indices of socio-demographic characteristics and reduced comparison of SES groups; we employed the Stata command “conindex” to calculate the p-value for testing if the concentration index is equal to zero. Because our outcomes were binary, the decomposition of the concentration index involved using a probit model with marginal effects for all socio-demographic explanatory variables. We only focused on how socio-demographic explanatory variables influenced healthcare inequalities, so we did not include other health variables and clinical factors and reduced the socio-demographic variables from the model.

• Ethics: Describe the measures taken to protect participants’ privacy and confidentiality. Also, did the participants receive any compensation?

Response: We have made revisions accordingly. (Lines 174-176, 179-182 and 338-345). Because we conducted the study when participants were attending the routine visits, there was no compensation given to them.

6. Results

• The wording and terminologies used in the description of the results need improvement.

Response: We have checked and made revisions accordingly. We have rewritten several parts of the results to make them clearer and easier to read.

• Descriptive results can be summarized as they are mostly repeating the information in tables.

Response: We have made revisions accordingly.

• Table 2: Exact p values should be presented rather than showing *p-value <0.05, ** p-value <0.01, and ***p-value <0.001 in the footnote.

Response: We have made revisions accordingly. (Table 1 and Table 2 )

7. Discussion

• Please refine the discussion according to the earlier comments and reviewer’s comments, focusing on this study's significant findings. It is also important to discuss the socio-economic burden of HIV care continuum on people living with HIV, particularly in the study setting context (e.g., free vs. charged services, other costs).

Response: Thank you for your suggestion. We have revised the discussion part and focussed on the significant findings. (Line 448-475)

8. References

• The reference list needs to be revised to be aligned with PLOS’s styles and requirements.

Response: We have revised the reference style as “Vancouver” based on PLOS’s requirements.

Journal Requirements:

Response: We have made revisions accordingly.

2. Please include additional information regarding the survey or questionnaire used in the study and ensure that you have provided sufficient details that others could replicate the analyses. For instance, if you developed a questionnaire as part of this study and it is not under a copyright more restrictive than CC-BY, please include a copy, in both the original language and English, as Supporting Information, or include a citation if it has been published previously.

Response: We have added accordingly.

Response: We have added and update accordingly.

Reviewers' comments:

1. Is the manuscript technically sound, and do the data support the conclusions?

Reviewer #1: Partly

2. Has the statistical analysis been performed appropriately and rigorously?

Reviewer #1: Yes

3. Have the authors made all data underlying the findings in their manuscript fully available?

Reviewer #1: No

4. Is the manuscript presented in an intelligible fashion and written in standard English?

Reviewer #1: No

Response: We have made revisions accordingly.

5. Review Comments to the Author

Reviewer #1: 

1) Many concentration indices were with non-significant p-values, but the authors still interpreted as if there were effect, which is inappropriate. The results of the manuscript should be summarized in a succinct manner with the significant and important results stressed.

Response: Thank you for your suggestion. We have revised the discussion part and focussed on the significant findings. (Line 448-475)

2) Table 3, it is unclear to me for the same variable why the absolute contribution and percentage of contribution to concentration index can have different signs?

Response: As we described in lines 282-285 and 293-297, the percentage of contribution equals (100*(βk ®xk / μ) Ck/ CIn). If the concentration index is negative, the absolute contribution can follow the concentration index sign as a negative value, but the contribution percentage was positive, so the positive contribution percentage increases the size of the inequality, a negative contribution percentage decreases the size of the inequality. 

3) Please ask for professional editing help with English language.

Response: Thank you. We have asked for professional editing help with the English.

---

## [Editor Report · Decision Letter 1]

7 Apr 2021

PONE-D-20-28608R1

Measuring and explaining inequality of continuous care for people living with HIV receiving antiretroviral therapy in Kunming, China

PLOS ONE

Dear Dr. Assanangkornchai,

Thank you for submitting your manuscript to PLOS ONE. After careful consideration, we feel that it has merit but does not fully meet PLOS ONE’s publication criteria as it currently stands. Therefore, we invite you to submit a revised version of the manuscript that addresses the points raised during the review process.

We look forward to receiving your revised manuscript.

Kind regards,

Siyan Yi, MD, MHSc, PhD

Academic Editor

PLOS ONE

Journal Requirements:

Additional Editor Comments:

We thank the authors for addressing the reviewers’ comments. The revised manuscript has been greatly improved. However, it still requires further attention, particularly in the Introduction and Methods. Below are some suggestions, that may not be exhaustive. The revised manuscript requires thorough proofreading to remove several minor grammatical errors, typos, terminology use, tenses, etc. Also, please ensure that your revised manuscript is prepared according to the journal’s styles and guidelines, particularly in the reference list. Please note that PLOS usually does not provide an opportunity for the authors to proofread their paper before the publication if it is accepted.

Title page:

Please use a consistent font and font size.

Abstract:

As commented earlier ‘people living with HIV’ should not be abbreviated or addressed as patients. This is applied across the paper, including keywords. See: https://www.unaids.org/sites/default/files/media_asset/2015_terminology_guidelines_en.pdfPlease refine the objective of the study, which is too long and difficult to read. The authors may consider breaking it down into two sentences. Also, ‘impact’ is an incorrect term in this context, given the study’s cross-sectional nature.Results:Line 28: what does ‘uptakes of retention’ in care mean?Is there any other ways to explain ‘non-significant pro-rich and pro-poor,’ that are easier to understand?Please make it clear if the last results (Lines 31-33) were statically significant.

Introduction

Overall, the introduction is well written but unnecessarily too long that should be condensed. Please try to limit it to approximately 1000 words (1-2 paragraphs for problem statement, 1-2 paragraphs on literature review of previous studies on inequality in access to HIV services and its associated factors, and 1 paragraph on rationales, and 1 paragraph on study’s objectives).Please also proofread it carefully after the revisions (e.g., on line 45, no need to write ART in full as it was already spelled out on line 43)

Materials and Methods

Line 150-151: It sounds a little bit awkward when saying, “Kunming city was chosen as the study area because it has the highest prevalence of HIV/AIDS in Yunnan.” Any rationale directly linked to Kunming? The prevalence of HIV also needs a reference to support. Please remove ‘AIDS’ from the highest prevalence of HIV/AIDS. As commented earlier, PLWH is not currently in use as recommended by UNAIDS.Eligibility criteria:Does HIV infection require a clinical diagnosis? Many people living with HIV do not present with any symptoms and are diagnosed only by serological tests – were they eligible for the study? Usually, getting registered to receive HIV services indicates that they live with HIV, which is clearly stated in criteria 3.Any criteria related to physical and mental health stability to participate in the interview?Lines 154, 157, 159: As commented earlier, numbers smaller than 10 should be spelled out in words in scientific and academic writing. This is applied to the whole document.Line 155: Please remove ‘,’ before ‘and.’Lines 156-161: The sentence is too long and should be broken down into 2-3 sentences.Line 161: Please replace ‘;’ by ‘.’Lines 165-169: The sentence is too long and should be broken down into 2 sentences.Line 184: Please spell out NFATP if it was used for the first time in the manuscript.Line 192: Guess ‘outcome’ should read ‘outcome variable.’Line 197: Not sure if the word ‘free’ before ‘ART manual’ is necessary. Is the manual applied only for ‘free’ ART? I do not find it necessary to keep saying ‘free ART’ (e.g., lines 198, 201) as it brings more confusion than help. The authors may just describe in the description of the study site that the HIV center where the study was conducted provides ART and other HIV services free of charges.Line 211: The use of ‘;’ before ‘and’ is incorrect.The NFATP only provides free CD4 and viral load testing once a year, and people living with HIV have to pay for the remaining tests. It is important to state whether the non-free tests were voluntary based on the clients’ ability to pay or compulsory as the test costs can be beyond their affordability. This may explain the relationships between inequality and socio-demographic characteristics, leaving alone other potential financial barriers (transport costs, busy working). I would suggest this point be discussed in the discussion section. This may also explain the low rate of viral load test presented in the results.Line 256: Please use past tense in the methods section (e.g., this study used…).Line 329: …using the Chi-square test.Line 352: Please present the median value with an interquartile range.Line 496: WHO was already defined in the introduction.Lines 497-498: the sentence “A study in Thailand suggested that simple and inexpensive monitoring of key biomarkers could be done at some time points” is not clear and should be further elaborated. What did the authors intend to tell – feasibility, efficacy, or else?Lines 523-524: A self-reporting measure can lead to both under- or over-reporting unless this study could prove that it was under-reporting.Line 556: Guess Kunming does not require ‘the.’Lines 556-559: Conclusions should summarize the key findings that respond to the research questions and objectives. The comparison of this study’s findings to the literature should be discussed in the discussion section.Lines 563-565: The sentence “Furthermore, need variables (i.e., age and sex) could not negatively contribute to all concentration indices to diminish the degree of inequalities” needs clarity as it is hard to understand.References are still inconsistent and not entirely aligned with PLOS’s style and requirements. Please ensure that they have been corrected before re-submission.

---

## [Author Response · Author response to Decision Letter 1]

21 Apr 2021

Rebuttal letter to Editor and Reviewer

PLOS One Manuscript number: PONE-D-20-28608

Measuring and explaining inequality of continuous care for people living with HIV receiving antiretroviral therapy in Kunming, China

Journal Requirements:

Response: As we condensed the introduction, we removed six references and revised the reference list.

Additional Editor Comments:

We thank the authors for addressing the reviewers’ comments. The revised manuscript has been greatly improved. However, it still requires further attention, particularly in the Introduction and Methods. Below are some suggestions, that may not be exhaustive. The revised manuscript requires thorough proofreading to remove several minor grammatical errors, typos, terminology use, tenses, etc. Also, please ensure that your revised manuscript is prepared according to the journal’s styles and guidelines, particularly in the reference list. Please note that PLOS usually does not provide an opportunity for the authors to proofread their paper before the publication if it is accepted.

Response: Thank you for your valuable comments. We have checked and made revisions accordingly, especially Introduction and Methods. We have checked and made revisions in the reference list based on the journal’s styles and guidelines.

Title page:

1. Please use a consistent font and font size.

Response: We have checked and made revisions accordingly.

Abstract:

2. As commented earlier ‘people living with HIV’ should not be abbreviated or addressed as patients. This is applied across the paper, including keywords. See: https://www.unaids.org/sites/default/files/media_asset/2015_terminology_guidelines_en.pdf

Response: We made revisions accordingly across the paper.

3. Please refine the objective of the study, which is too long and difficult to read. The authors may consider breaking it down into two sentences. Also, ‘impact’ is an incorrect term in this context, given the study’s cross-sectional nature.

Response: We have made revisions accordingly. (Page 2 lines 15-19: We have split the objectives into two sentences. Line 19: We changed “impact” to “contributions”.)

4. Results: 

• Line 28: what does ‘uptakes of retention’ in care mean?

Response: We have modified this sentence accordingly. (Page 2 line 28: We removed “Uptakes of.”)

• Is there any other ways to explain ‘non-significant pro-rich and pro-poor,’ that are easier to understand?

Response: We have revised this as follows: (Page2 lines 29: We revised “non-significantly pro-rich” to “non-significant in favor of the rich” And lines 30-31: we revised “nonsignificant pro-poor” to “higher among the poor.”)

• Please make it clear if the last results (Lines 31-33) were statistically significant.

Response: We calculated contributions of socioeconomic status to inequalities based on equation (5), finally we did not check for statistical significance.

Introduction

5. Overall, the introduction is well written but unnecessarily too long that should be condensed. Please try to limit it to approximately 1000 words (1-2 paragraphs for problem statement, 1-2 paragraphs on literature review of previous studies on inequality in access to HIV services and its associated factors, and 1 paragraph on rationales, and 1 paragraph on study’s objectives).

Response: Thank you for your comments. We have checked revised the introduction accordingly.(Pages 3-6)

6. Please also proofread it carefully after the revisions (e.g., on line 45, no need to write ART in full as it was already spelled out on line 43)

Response: Checked and revised accordingly.

Materials and Methods

7. Line 150-151: It sounds a little bit awkward when saying, “Kunming city was chosen as the study area because it has the highest prevalence of HIV/AIDS in Yunnan.” Any rationale directly linked to Kunming? The prevalence of HIV also needs a reference to support. Please remove ‘AIDS’ from the highest prevalence of HIV/AIDS. As commented earlier, PLWH is not currently in use as recommended by UNAIDS.

Response: Revised accordingly.

Lines 116-117: As we have no rationale directly linked to Kunming, thus we have removed the sentence referring to Kunming city being chosen due to its high prevalence of HIV/AIDS and added: “Kunming is the capital city of Yunnan province.” We have also removed “AIDS” accordingly.

8. Eligibility criteria:

• Does HIV infection require a clinical diagnosis? Many people living with HIV do not present with any symptoms and are diagnosed only by serological tests – were they eligible for the study? Usually, getting registered to receive HIV services indicates that they live with HIV, which is clearly stated in criteria 3.

Response: Thank you for your suggestion. We did not require a clinical diagnosis for HIV infection. We have therefore removed this inclusion criteria.

• Any criteria related to physical and mental health stability to participate in the interview?

Response: Yes.

Page 6 line 122-124: We added the following criteria: “individuals were able to present themselves on the day of the interview and sufficiently physically and mentally stable to provide verbal consent to participate in the study” to participate in the study.

• Lines 154, 157, 159: As commented earlier, numbers smaller than 10 should be spelled out in words in scientific and academic writing. This is applied to the whole document.

Response: Revised accordingly.

• Line 155: Please remove ‘,’ before ‘and.’

Response: Done.

• Lines 156-161: The sentence is too long and should be broken down into 2-3 sentences.

Response: Done. (Pages 6-7 lines 124-130)

• Line 161: Please replace ‘;’ by. ‘’

Response: Done. (Page 7 line 130)

• Lines 165-169: The sentence is too long and should be broken down into 2 sentences.

Response: Done. (Page 7 lines 134-139)

9. Line 184: Please spell out NFATP if it was used for the first time in the manuscript.

Response: NFATP was spelled out in the Introduction section. (Page 4 line 66)

10. Line 192: Guess ‘outcome’ should read ‘outcome variable.’

Response: Yes - revised accordingly. (Page 8 line 161)

11. Line 197: Not sure if the word ‘free’ before ‘ART manual’ is necessary. Is the manual applied only for ‘free’ ART? I do not find it necessary to keep saying ‘free ART’ (e.g., lines 198, 201) as it brings more confusion than help. The authors may just describe in the description of the study site that the HIV center where the study was conducted provides ART and other HIV services free of charges.

Response: Thank you. On page 8 lines 167-171: The manual applied only for free-ART, the HIV center provides free ART, and other services are not free of charge. We have the manual to guide the free-ART services, listed in reference 20, and the Chinese national guideline to guide the national treatment and services for people living HIV, listed in reference 21.

12. Line 211: The use of ‘;’ before ‘and’ is incorrect.

Response: Thank you. Revised accordingly. (Page 9 line 181)

13. The NFATP only provides free CD4 and viral load testing once a year, and people living with HIV have to pay for the remaining tests. It is important to state whether the non-free tests were voluntary based on the clients’ ability to pay or compulsory as the test costs can be beyond their affordability. This may explain the relationships between inequality and socio-demographic characteristics, leaving alone other potential financial barriers (transport costs, busy working). I would suggest this point be discussed in the discussion section. This may also explain the low rate of viral load test presented in the results.

Response: Thank you for your valuable comments. We have made some revisions accordingly. (Page 9 lines 184, 188-189: the non-free tests were voluntary based on the client’s ability to pay, and test costs can be beyond their affordability. Page 24 lines 479-482: We added this point into the Discussion section.)

14. Line 256: Please use past tense in the methods section (e.g., this study used…).

Response: Done. (Page11 line 228)

15. Line 329: …using the Chi-square test.

Response: Thank you. (Page 14 line 303: SES groups were calculated and compared using the chi-square test.)

16. Line 352: Please present the median value with an interquartile range.

Response: Done. (Page 15 line 326: median age of 41.0 years (interquartile range, 32.0 to 48.0 years))

17. Line 496: WHO was already defined in the introduction’.

Response: Thanks and revised accordingly. (Page24 line 471)

18. Lines 497-498: the sentence “A study in Thailand suggested that simple and inexpensive monitoring of key biomarkers could be done at some time points” is not clear and should be further elaborated. What did the authors intend to tell – feasibility, efficacy, or else?

Response:We meant feasibility. (Page 24 lines 472-473: A study in Thailand demonstrated that the simple and inexpensive monitoring of key biomarkers was feasible at some time points)

Lines 523-524: A self-reporting measure can lead to both under- or over-reporting unless this study could prove that it was under-reporting.

Response: We cannot prove either way. (Page 26 lines 499-500: the household assets and conditions were self-reported, which might result in both under- or over-reporting of participants’ SES.)

20. Line 556: Guess Kunming does not require ‘the.’

Response: Yes, of course. Corrected.(Page 27 line 534)

21. Lines 556-559: Conclusions should summarize the key findings that respond to the research questions and objectives. The comparison of this study’s findings to the literature should be discussed in the discussion section.

Response: Thank you. Please see page 27 lines 534-537.

Lines 563-565: The sentence “Furthermore, need variables (i.e., age and sex) could not negatively contribute to all concentration indices to diminish the degree of inequalities” needs clarity as it is hard to understand.

Response: We have revised this as follows: (Page27 lines 541-544: Between the two need variables (age and sex), age contributed more to the inequalities in the utilization of all HIV continuous care. This implies that the degree of such inequalities can be reduced should comprehensive care be provided equally to people living with HIV of all age groups.)

23. References are still inconsistent and not entirely aligned with PLOS’s style and requirements. Please ensure that they have been corrected before re-submission.

Response: We have checked the references carefully and revised them accordingly. (Pages 29-33)

---

## [Editor Report · Decision Letter 2]

23 Apr 2021

Measuring and explaining inequality of continuous care for people living with HIV receiving antiretroviral therapy in Kunming, China

PONE-D-20-28608R2

Dear Dr. Assanangkornchai,

We’re pleased to inform you that your manuscript has been judged scientifically suitable for publication and will be formally accepted for publication once it meets all outstanding technical requirements.

Kind regards,

Siyan Yi, MD, MHSc, PhD

Academic Editor

PLOS ONE
---

## [Editor Report · Acceptance letter]

30 Apr 2021

PONE-D-20-28608R2 

Measuring and explaining inequality of continuous care for people living with HIV receiving antiretroviral therapy in Kunming, China

Dear Dr. Assanangkornchai:

I'm pleased to inform you that your manuscript has been deemed suitable for publication in PLOS ONE. Congratulations! Your manuscript is now with our production department. 

Kind regards, 

on behalf of

Dr. Siyan Yi 

Academic Editor

PLOS ONE